# Moving towards Digitising COVID-19 Vaccination Certificate: A Systematic Review of Literature

**DOI:** 10.3390/vaccines10122040

**Published:** 2022-11-29

**Authors:** Jonathan Kissi, Emmanuel Kusi Achampong, Nathan Kumasenu Mensah, Caleb Annobil, Jessica Naa Lamptey

**Affiliations:** 1Department of Health Information Management, School of Allied Health Sciences, University of Cape Coast, Cape Coast, Ghana; 2Department of Medical Education and ICT, School of Medical Sciences, University of Cape Coast, Cape Coast, Ghana

**Keywords:** COVID-19 passport, travel pass, digital COVID-19 vaccination certificate, paper-based COVID-19 vaccination certificate, COVID-19 passport policies

## Abstract

The coronavirus pandemic is the greatest crisis of our time, having claimed over 2 million lives and shocking the global economy. Scientists and governments have suggested the idea of a digital COVID-19 certificate, to identify vaccinated persons easily. This paper assesses the positions of stakeholders on COVID-19 vaccination certificates, their presentation, and their importance. Preferred Reporting Items for Systematic Reviews and Meta-Analyses (PRISMA) was applied in this study. Search terms with Boolean and/or operators were combined to increase relevant results. Four large digital databases were used for the search. Inclusion and exclusion criteria were used to screen 298 collated studies. Two reviewers independently assessed search results, extracted data, and assessed the quality of the included studies. It is essential to re-examine digital COVID-19 vaccination certificates, considering their benefits, such as real-time detection of fake vaccination certificates and identifying and mapping non-vaccinated areas for strategic vaccination planning. The use of a single electronic platform globally will ease verification processes while bringing economies back to their feet. Digital COVID-19 vaccination certificates may provide balance in this pandemic era. With digital COVID-19 certificate exceeding documentation purposes, it is important to recognise factors such as global economy and human rights, boosting free movements of persons.

## 1. Introduction

Global economy and health have been affected since the outbreak of acute community-based atypical pneumonia of unknown aetiology was reported in Wuhan, capital of Hubei province in China, in December 2019 [1]. Scores of researchers observe that global initiatives have been established for the deployment of vaccines to tackle the COVID-19 virus [2,3,4,5]. A vaccination certificate is issued out to any individual who is vaccinated against the virus. This vaccination certificate serves as proof to grant the holder access to public events, air travel, etc., without being restricted from accessing facilities in these areas [6]. A study by Hernández-Ramos et al. [7] shows that COVID-19 certificates have been given to vaccinated persons to ease restrictions imposed by governments. Although COVID-19 vaccination certificates can help ease restrictions, the processes involved in obtaining and using them should not be seen as agents of promoting the spread of the virus. Failure to focus on security systems to store patients’ information may raise data privacy concerns that can discourage many from partaking in the vaccination process [8].

The work of Schlagenhauf et al. [9] discusses whether COVID-19 vaccination certificates should be digitalised or in paper format. Should the COVID-19 vaccination certificate be an enhanced version of the International Certificate of Vaccination or Prophylaxis, potentially in the form of an electronic record? Another studies by Kowalewski et al. [10] proposes that digital tools play a vital role in mitigating the spread of COVID-19. Digitalising COVID-19 vaccination certificates can minimise physical contact during verification of vaccination status at public events and local and international airports. However, an online representative study conducted in Germany on a sample of 599 participants assessing user perception of vaccination certificates indicated that the majority of participants favoured paper-based COVID-19 vaccination certificates over digital COVID-19 vaccination certificates [10]. Obligatory COVID-19 certification (showing recent negative test, proof of recovery, or vaccination) has been tabled in some countries [11]. As part of global efforts to enhance the vaccination rate, scientists and governments have suggested the idea of COVID-19 immunity certificates. The COVID-19 immunity certificate is for those fully vaccinated or recovered from the coronavirus disease. The COVID-19 immunity certificate is an incentive to drive public interest in vaccination campaigns [12].

Vaccination is gaining global ground. The idea of a COVID-19 vaccination certificate is gaining popularity. Hence, there is the need for international health bodies (such as the World Health Organisation) to be specific as to whether COVID-19 vaccination certificates should be digitalised or paper-based [13]. Artificial intelligence can be applied to detect fake COVID-19 vaccination certificates if the certificate is in a digital format. Furthermore, artificial intelligence can help identify and map non-vaccinated regions for strategic planning if digital COVID-19 vaccination certification systems are deployed. Furthermore, through the application of artificial intelligence, digital COVID-19 vaccination certification systems can organise migration patterns of migrants, based on data stored in the verification application, and assist in contact tracing. Blockchain (BC) technology, which is an immutable, transparent, and decentralised technology that provides a secured way of sending information, may ensure patients’ personal data confidentiality [14]. Internet of Things and BC technology can be used to remotely access COVID-19 vaccination certificates. The paper-based vaccination certificate mostly serves “documenting” and “manual verification” purposes [15]. Although the European Union named its COVID-19 vaccination certificate the “EU Digital COVID Certificate”, some member states still combine digital COVID-19 vaccination certificates with paper-based COVID-19 vaccination certificates [16]. This paper assesses the positions of stakeholders on COVID-19 vaccination certificates and evaluates the formats stakeholders propose a COVID-19 vaccination certificate must take. The study also expounds on the importance of digitalising COVID-19 vaccination certificates.

## 2. Materials and Methods

### 2.1. Identification: Data Sources and Search Strategy

Keywords were defined to obtain relevant kinds of literature that address the learning objective. Search keywords were created to ensure the quality of the reviews. This was done by combining search terms with Boolean and/or operators to increase relevant results. With constraints on the search, Medical Subject Headings (MeSH) were used in the search. Keywords for this review were: COVID-19; Travel Pass; COVID-19 Vaccination Certificate; Digital COVID-19 Vaccination Certificate; Paper-based COVID-19 Vaccination Certificate; COVID-19 Passport Policies. Four digital databases were used for the search, and they were: ProQuest, PubMed, Sage Pub, and Science Direct (see Table 1). The rationale behind selecting these search engines was that they suit the study areas of this review, which are health/medical science and information systems. ProQuest, PubMed, Sage Pub, and Science Direct have a filter feature to screen literature based on year of publication, type of literature, study area, etc. The study title and keywords were used to search for related literature. The plan was to collate all results for each keyword search. The search was conducted from 23 October 2021 to 3 January 2022. A total of 298 studies were collated at the end of search. Two reviewers independently assessed search results, extracted data, and assessed the quality of the included studies.

### 2.2. Screening: Selecting Studies Based on Listed Criteria

The following inclusion criteria were used to screen the 298 collated studies. The paper needs to: (1) be written in English; (2) be a journal paper; (3) address review objectives; (4) be related to specified keywords. Duplicated papers and papers not in full text were excluded. This is depicted in Table 2.

Preferred Reporting Items for Systematic Reviews and Meta-Analyses (PRISMA) was applied in this study. Streamlining literature availability and finding best-fit literature for this review was the reason for using PRISMA flow diagram. This is shown in Figure 1.

## 3. Results

Out of the two hundred and seventy-one (271) excluded papers, twenty-three (23) papers, representing 9%, were excluded for not been written in English. There were fifty-nine (59) duplicates excluded, representing 22%. Sixty-nine (69) papers, representing 25%, were excluded because they were not journal papers. A total of one hundred and twenty (120) papers, representing 44%, were excluded for not relating to the subject matter. This is depicted in Figure 1.

Twenty-seven (27) papers were included in the qualitative synthesis (see Table 3). The inclusions consisted of three (3) articles, representing 11%, used for introduction to the study synthesis (see Table 4); three (3) articles, representing 11%, on the background of COVID-19 (see Table 5); four (4) articles, representing 15%, on the history of vaccination certificates (see Table 6); eleven (11) articles, representing 41%, that expounded on the positions of stakeholders on COVID-19 vaccination certificates (see Table 7); three (3) articles, representing 11%, that elaborated on the advantages and disadvantages of digital and paper-based COVID-19 vaccination certificates (see Table 8); and three (3) papers, representing 11%, on privacy concern issues around digital COVID-19 vaccination certificates (see Table 9).

Papers included in the review were from seventeen (17) different countries. 

Figure 2 shows the distribution based on countries.

## 4. Discussion

The review explores the positions of various stakeholders (the World Health Organisation, the International Civil Aviation Organisation, continental unions, countries, and global citizens) on COVID-19 vaccination certificates and evaluates the formats stakeholders propose a COVID-19 vaccination certificate must take. The study also expounds on the importance of digitalising COVID-19 vaccination certificates.

### 4.1. The Positions of Stakeholders on COVID-19 Vaccination Certificates

The World Health Organisation (WHO) suggests that COVID-19 vaccination requirements for international travels should not be implemented [22]. This may be a public health decision to limit the spread of the virus by limiting physical contacts that COVID-19 vaccination requirements introduce. The aviation industry was one of the sectors that took major hits as a result of the coronavirus pandemic [19]. The International Civil Aviation Organisation (ICAO) has indicated its intent to accept proof of COVID-19 immunisation in the form of a WHO yellow card [4]. Is this an economic decision to bring the aviation sector back to its feet or a decision to balance global public health and economy? At a regional level, the European Union (EU) has developed its own EU Digital COVID-19 certificate, which has been accepted by all member states [16]. Although the name of the European Union’s certificate is the EU Digital COVID-19 Certificate, its implementation is a blend of digital and paper-based certificates. At a national level, while some countries such as Lithuania use strict measures to ensure high vaccination rates, others such as Poland are adapting the voluntary option [26]. Compulsory COVID-19 certification has been initiated in some countries [11]. The Chilean government has initiated and implemented a “release passport” policy, which is similar to an immunity passport, a position opposed by the World Health Organisation [3]. Bhutan “encourages” visitors to be vaccinated but does not mandate them to be vaccinated [13]. Other studies have been conducted to assess the perception of citizens around the world of COVID-19 vaccination certificates [24,25]. A study conducted in the United Kingdom showed that most people were inclined towards accepting the COVID-19 vaccine because of the COVID-19 vaccination certificate [25]. Another study that assessed behavioural responses of participants to COVID-19 health certification indicated that public attitudes towards the use of immunity certificates for international travels were favourable. However, the public opposed the idea of using immunity certificates for work and other activities [24].

From international, regional, national, and citizenry levels, all stakeholders agreed on the need for COVID-19 vaccination programmes [4,5,22]. While some stakeholders suggested that COVID-19 certificates should not be used as evidence for international travels, others made known their intent to accept proof of COVID-19 vaccination for international travels [4,22]. This is a policy conflict between stance of public health and that of global economy [12]. Striking a balance between public health and bringing the global economy back to its feet is the solution. Digitalising COVID-19 vaccination certificates will reduce public health risks associated with vaccination certification policies. Bringing the global economy back to its feet must not be done in a hurried manner that may escalate the COVID-19 virus [19,20].

While some countries are mandating COVID-19 vaccination certificates, others suggest it must be a voluntary decision [3,11,13,26]. This is a conflict between human rights and public health. Should the rights of people be infringed upon in a quest to implement public health policies? A COVID-19 vaccination certificate is now a requirement to attend public events and to travel locally and internationally [24,25]. A COVID-19 vaccination certificate is also required to allow people to work in some jurisdictions [26]. How can a policy strike a balance between public health and human rights?

### 4.2. Policy Recommendation on the Positions of Stakeholders on COVID-19 Vaccination Certificates

Score of researchers has postulated that policies from the World Health Organisation recognising factors such as global economy and human rights while showing a framework to implement a COVID-19 vaccination certification programme globally will help to promote uniformity in COVID-19 vaccination certificate programmes and promote the onboarding of all stakeholders [3,4,9,15,22,23,24,25].

### 4.3. The Format COVID-19 Vaccination Certificate Must Take

The EU Digital COVID-19 Certificate introduced by the European Union includes three certificate types: test certificates, vaccination certificates, and recovery certificates [23]. While issues of falsification or counterfeit COVID-19 vaccine certificates surround the paper-based approach, privacy issues also surround the digital approach [9]. A web-based study evaluated user perception of COVID-19 vaccination certificates. Two paper-based variants and three digital app variants were used. Digital and paper-based vaccine certificates were used for documentation purposes only [assumed]. The results showed that paper-based vaccination certificates were generally favoured over digital COVID-19 vaccination certificates [10]. In addition, benefits of digital COVID-19 vaccination certificates and paper-based COVID-19 vaccination certificates should not be limited to documentation purposes only, as cited in [10]. Digital COVID-19 vaccination certificates can be used to execute various functions. By applying artificial intelligence, fake COVID-19 vaccination certificates can be easily detected, non-vaccinated areas can be identified and mapped for strategic vaccination planning, and contact tracing can be improved. Furthermore, Internet of Medical Things (IoMT) and blockchain technology can be applied in areas with remote access to COVID-19 vaccination certificates, ensuring data privacy [15]. The study conducted by Mbunge et al. [15] highlights the need for global adaptation towards use of digital COVID-19 vaccination certificates and uniform global implementation of digital COVID-19 vaccination certificates.

### 4.4. Policy Recommendation on Format of COVID-19 Vaccination Certificates

According to Sharif et al. [5], the use of a single electronic platform globally will ease verification processes while bringing economies back to their feet. A policy that initiates global digitalisation of COVID-19 vaccination certificates while highlighting technologies to mitigate data privacy issues associated with digitalised COVID-19 vaccination certificates will enhance stakeholders’ buy-in.

### 4.5. Addressing Challenges Associated with Digital COVID-19 Vaccination Certificates: A Focus on Data Privacy

Data privacy is the main challenge that hinders people from buying into the idea of digital COVID-19 vaccination certificates [25]. In addressing privacy challenges, blockchain technology can be a major solution [7]. The immutability, transparency, and decentralisation characteristics of blockchain technology provide a trustworthy and secured way of sharing confidential and personal data [14]. One vulnerability of blockchain technology is apparent when resolution of an on-chain mapping of an identifier is specified to a key in public-permissioned blockchain systems, which results in redirection to central servers [27]. Encrypting on-chain mapping processes of an identifier could mitigate this limitation.

### 4.6. Policy Recommendation Addressing Challenges Associated with Digital COVID-19 Vaccination Certificates: Focus on Data Privacy

A policy recommendation inferred from Foy et al. [27] stipulates that resolution of on-chain mapping of an identifier to a key in public-permissioned blockchain systems must not be specified, as it will redirect results to central servers, exposing data of vaccinated persons.

## 5. Conclusions

Digital COVID-19 vaccination certificates may provide balance in this pandemic era [16]. Studies postulate them as negotiating the thin line between public health and bringing the global economy back to its feet [19,20]. With use of digital COVID-19 vaccination certificates exceeding documentation purposes, recognising factors such as global economy and human rights while showing a framework to implement uniform global digital COVID-19 vaccination certification will mitigate COVID-19 spread while boosting international travels [14]. It is essential to re-examine digital COVID-19 vaccination certificates, taking into account their benefits, such as: real-time detection of fake COVID-19 vaccination certificates, identifying and mapping non-vaccinated areas for strategic vaccination planning, aiding in contact tracing, and supporting areas with remote access to COVID-19 vaccination certificates [15]. In addressing privacy challenges, blockchain technology can be a major solution [7]. Several studies expound on the stances of stakeholders on COVID-19 vaccination certificates and the format they must take as has been stipulated in this review study. Hence, there is a need for a global policy in that direction by the World Health Organisation to promote uniformity in its implementation.

Quintessential of academic studies, there were some limitations to this study. For instance, the literature search was conducted in only four databases. This may create an impression that studies outside the selected databases were ignored. Some studies with relevant information were disregarded after screening of downloaded studies using inclusive and exclusive criteria. To some extent, methodological limitations of included studies that utilised primary data may have influenced findings of this review. Furthermore, generalisability of findings of this study may be limited, because included studies did not clearly elaborate on how participants were selected. With respect to secondary research-based included studies, limitations regarding the process of sampling were disclosed. This may also limit findings of this research. The included articles in this study did not equally cover all the geographical areas of the world but were limited to only seventeen countries, as shown in Figure 2. It must be reemphasised that only English-language articles were included. This also limits the generalisability of the findings to other contexts. This review is a contribution to the strategies and recommendations for implementing COVID-19 vaccination certificates based on the included databases. Further primary-data-based studies must investigate how paper-based vaccination certificates may have contributed to the spread of COVID-19 and projections into the future.

## Figures and Tables

**Figure 1 vaccines-10-02040-f001:**
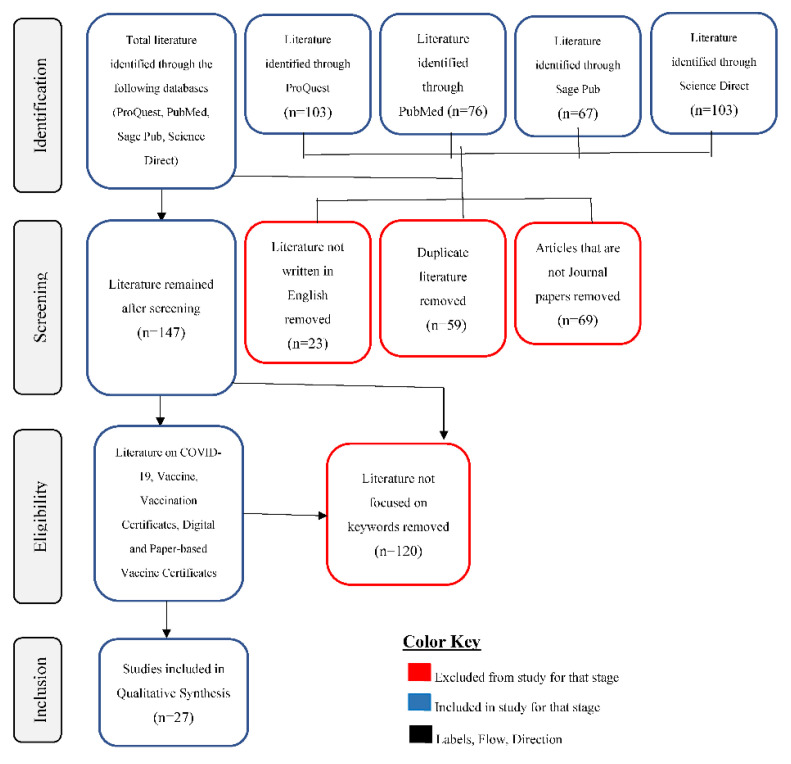
PRISMA flow for process of article inclusion and exclusion.

**Figure 2 vaccines-10-02040-f002:**
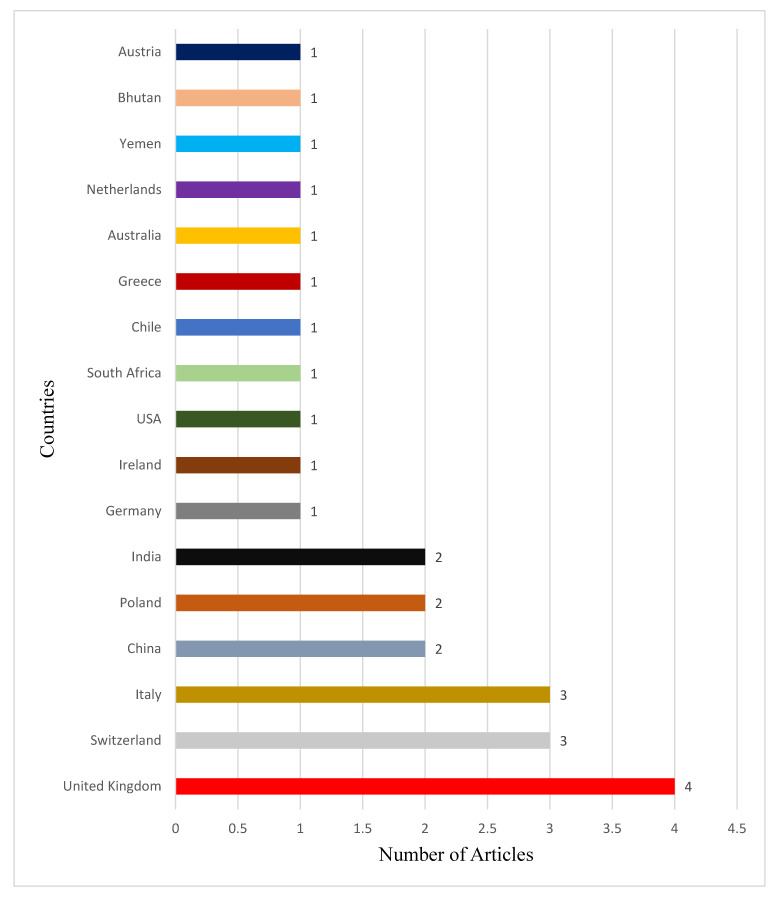
Distribution of countries based on their respective number of articles. Summary characteristics of studies included.

**Table 1 vaccines-10-02040-t001:** Databases and number of studies used in this review.

	Number of Studies Used	Percentage of Studies Used
ProQuest	11	41%
PubMed	9	33%
Sage Pub	3	11%
Science Direct	4	15%
Total	27	100%

**Table 2 vaccines-10-02040-t002:** Inclusion and exclusion of studies based on criteria.

	Decision
Papers written in English	Include
Papers addressing learning questions and study objective	Include
Papers related to specified keywords	Include
Duplicated papers in collated documents	Exclude
Papers that are not in journals	Exclude
Papers not in full text	Exclude

**Table 3 vaccines-10-02040-t003:** Results of Included Articles.

	Subject Matter	Number of Articles	Percentages of Articles
4.0	Introduction to Study Analysis	3	11%
5.0	Background of COVID-19	3	11%
6.0	History of Vaccination Certificates	4	15%
7.0	Positions of Stakeholders on COVID-19 Vaccination Certificates	11	41%
8.0	Digital and Paper-based COVID-19 Vaccination Certificates: The Advantages and Disadvantages	3	11%
9.0	Addressing Issues of Privacy Concerns for Digital COVID-19 Vaccine Certificates	3	11%

**Table 4 vaccines-10-02040-t004:** Introduction to study analysis.

Year of Publication	Citation	Title	Study Location	Key Findings
2021	World Health Organisation, [1]	Looking back at a year that changed the world WHO’s Response to COVID-19	Geneva, Switzerland.	The SARS-CoV-2 virus pandemic is the greatest crisis of our time, having claimed over 2 million lives and shocking the global economy.
2020	Nueangnong et al. [17]	The 2020s world deadliest pandemic: Corona Virus (COVID-19) and International Medical Law (IML)	Department of Accounting, Hodeidah University, Hodeidah, Yemen.	In order to contain the spread of the SARS-CoV-2 virus, the World Health Organisation proposed that governments and their citizens avoiding touching their eyes, nose, and mouth with their hands. People were required to limit physical contacts.
2021	Martin, [18]	Aadhaar in a Box? Legitimizing Digital Identity in Times of Crises.	Tilburg University, The Netherlands.	There has been a drive politically to develop digital identity systems, with financial and technical support from international development stakeholders, in this SARS-CoV-2 virus pandemic era.

**Table 5 vaccines-10-02040-t005:** Background of COVID-19.

Year of Publication	Citation	Title	Study Location	Key Findings	Critique
2021	To et al. [2]	Lessons learned 1 year after SARS-CoV-2 emergence leading to COVID-19 Pandemic	State Key Laboratory of Emerging Infectious Diseases, The University of Hong Kong, Pokfulam, Hong Kong, China.	In December 2019 in Wuhan, the capital of Hubei province in central China, an outbreak of acute community-acquired atypical pneumonia of unknown aetiology was first recorded. Other studies suggest that 45% of cases recorded before 1 January 2020 had not correlation with the market in Wuhan.	The study could not provide enough information on how COVID-19 spread between cities by air travel, resulting in global spread.
2021	Sotis et al. [19]	COVID-19 Vaccine Passport and International Traveling: The Combined Effect of Two Nudges on Americans’ Support for the Pass	London School of Economics, London, United Kingdom.	International travel was one sector that was severely hit by the COVID-19 pandemic.	The study only highlighted the need to introduce a COVID-19 vaccine certificate, to bring the global economy back to its feet. However, the study was silent on how to balance public health with this decision.
2021	Paul et al. [20]	Austria’s Digital Vaccination Registry: Stakeholder Views and Implications for Governance	University of Vienna, Vienna, Austria.	To mitigate the spread of COVID-19, healthcare institutions attribute potential for enhancing efficiency and quality of provided health services to eHealth.	The study did not highlight how challenges associated with digital vaccination registries can be mitigated.

**Table 6 vaccines-10-02040-t006:** History of vaccination certificates.

Year of Publication	Citation	Title	Study Location	Key Findings	Critique
2021	Khan et al. [6]	Conflicting attitudes: Analysing social media data to understand the early discourse on COVID-19 passports.	Social Media Analytics Research Team (SMART) Lab, Scripps College of Communication, Ohio University, USA.	A vaccine passport is a paper or digital document that serves as proof of COVID-19 vaccine, permitting entry to sporting events, air travel, and public venues.	A vaccine passport is not proof of coronavirus vaccine but proof of taking a COVID-19 vaccine shot.
2021	Hu et al. [12]	Passport to a Mighty Nation: Exploring Sociocultural Foundation of Chinese Public’s Attitude to COVID-19 Vaccine Certificates	School of Communication, Soochow University, Suzhou, China.	Scientists and governments have suggested the idea of a COVID-19 immunity passport to improve vaccination rate globally. The immunity passport has been proposed for people vaccinated against the coronavirus or people who have recovered from the COVID-19 disease.	The rationale behind introducing the immunity passport was not well defined. This is because clearly defining the rationale would have led to a critical look at how to verify vaccinated people with minimum contact.
2021	Riva et al. [21]	COVID-19 health passes: lessons from the past	School of Medicine and Surgery, University of Milano-Bicocca, Monza, Italy.	All travellers in Northern Italy during the Renaissance who wished to vacate the city were mandated to obtain a health passport that certified that the individual came from a place free of plague and was safe. This health passport was initially hand-written. In a quest to prevent fraud, the health passport was later printed on small slips of paper to be completed by hand.	Although the study provided extensive information on paper-based vaccination certificates, it failed to equally provide some background on the emergence of digital vaccination certificates.
2021	Choudhary et al. [8]	Vaccination certificate: An initiative to mitigate COVID-19 waves in India?	Central Agricultural University, Selesih, Aizawl, Mizoram, India.	This is not the first time vaccination certificates have been introduced. Yellow fever and meningococcus require proof of vaccination.	Although there have been other vaccination certificates, the discussion must extend to the nature of diseases that require vaccination certificates.

**Table 7 vaccines-10-02040-t007:** Positions of stakeholders on COVID-19 vaccination certificates.

Year of Publication	Citation	Title	Study Location	Key Findings	Critique
2021	Pavli and Maltezou, [22]	COVID-19 vaccine passport for safe resumption of travel	Department of Travel Medicine, National Public Health Organization, Athens, Greece.	The World Health Organisation’s position is that conveyance operators and national authorities should not introduce requirements of Coronavirus vaccinations for international travels.	The reason behind the stance of the World Health Organisation (WHO) has not been highlighted.
2021	Petersen et al. [4]	COVID-19 vaccines under the International Health Regulations–We must use the WHO International Certificate of Vaccination or Prophylaxis	European Society for Clinical Microbiology and Infectious Diseases, ESCMID, Basel, Switzerland.	The International Civil Aviation Organisation (ICAO) are willing to accept proof of COVID-19 immunisation in the form of a WHO yellow card. A digital “Yellow Card” is preferred, but there is no WHO-endorsed platform available.	The study did not indicate if the ICAO was focused on bringing the global economy back to life, since their stance is the opposite of that of the WHO.
2021	Sharif et al. [5]	A pragmatic Approach to COVID-19 Vaccine Passport	Institute for Physical Activity and Nutrition, Deakin University, Bunwood, Victoria, Australia.	Most governments across the world are developing strategies to protect public health while returning life to almost normal. There have been conversations around deploying COVID-19 vaccine passports to serve as tools for easing lockdowns and strict travel restrictions.	Are governments’ decisions to create balance between public health and global economy in line with WHO recommendations?
2020	Fraser, [3]	Chile Plans Controversial COVID-19 Certificates	Chile.	The “release certificate”, which is almost like an “immunity passport”, is an idea that is not supported by the WHO but has been implemented by the Chilean Government to allow people who have recovered from COVID-19 to serve their country with less chance of been reinfected or infecting others.	The study could not explicitly recommend solutions to the Chilean government that could help bring back their economy while ensuring the public health framework was followed.
2021	Raciborski et al. [16]	Factors Associated with a Lack of Willingness to Vaccinate against COVID-19 in Poland: A 2021 Nationwide Cross-Sectional Survey.	Department of Prevention of Environmental Hazards and Allergology, Medical University of Warsaw, Warsaw, Poland.	The European Union has launched the EU Digital COVID-19 Certificate, which covers test, vaccination, and recovery and is in full force in all member states.	Although the name is “EU Digital COVID-19 Certificate”, members of the EU receive both digital and paper-based COVID-19 certificates.
2021	Vergallo et al. [23]	Does the EU COVID Digital Certificate Strike a Reasonable Balance between Mobility Needs and Public Health?	Department of Anatomical, Histological, Forensic and Orthopedic Sciences, Sapienza University of Rome, Italy.	The proposed EU Digital COVID Certificate includes three certificate types: test certificates, vaccination certificates, and recovery certificates.	The stance of including a recovery certificate opposes the stance of the World Health Organisation.
2021	Drury et al. [24]	Behavioural responses to COVID-19 health certification: a rapid review	School of Psychology, University of Sussex, Brighton, United Kingdom.	Public attitudes towards the use of immunity certificates for international travels were favourable. However, the public opposed the idea of using immunity certificates for work and other activities.	The study did not extensively highlight reasons why the public favoured immunity certificates for travels but rejected the idea of using the same certificate for work and other purposes.
2021	Mills and Rüttenauer, [11]	The effect of mandatory COVID-19 certificates on vaccine uptake: synthetic-control modelling of six countries.	Leverhulme Centre for Demographic Science, Nuffield College and Pandemic Sciences Centre, University of Oxford, Oxford, United Kingdom.	Compulsory COVID-19 certification has been initiated in some countries such as France, Germany, and Denmark.	There was no clear indication of whether the implementation of mandatory certification was fully digital or a mix of digital and paper-based.
2021	Tamang et al. [13]	Control of travel-related COVID-19 in Bhutan	Central Regional Referral Hospital, Gelegphu, Bhutan.	Bhutan “encourages” visitors to be vaccinated but does not mandate them to be vaccinated.	There was no conclusion indicating if encouraging visitors to be vaccinated resulted in high vaccination levels than mandating they be vaccinated.
2021	de Figueiredo et al. [25]	The Potential Impact of Vaccine Passports on Inclination to Accept COVID-19 Vaccinations in the United Kingdom: Evidence from a Large Cross-Sectional Survey and Modeling Study	Department of Infectious Disease Epidemiology, London School of Hygiene and Tropical Medicine, London, United Kingdom.	Based on a study conducted in the UK, most people are inclined towards accepting vaccination due to vaccination card.	The study was silent on the need for a global policy on COVID-19 vaccine cards based on the interest of people.
2021	Walkowiak et al. [26]	COVID-19 Passport as a Factor Determining the Success of National Vaccination Campaigns: Does It Work? The Case of Lithuania vs. Poland	Department of Preventive Medicine, Poznan University of Medical Sciences, Poznan, Poland.	Lithuania recorded a high vaccination rate compared to Poland because the country enforced strict vaccination rules. Unvaccinated persons were not even allowed in supermarkets.	The study also did not highlight the fraction of people with paper-based certificates compared to digital certificates.

**Table 8 vaccines-10-02040-t008:** Digital and paper-based COVID-19 vaccination certificates: the advantages and disadvantages.

Year of Publication	Citation	Title	Study Location	Key Findings	Critique
2021	Schlagenhauf et al. [9]	Variants, vaccines and vaccination passports: Challenges and chances for travel medicine in 2021	University of Zürich Centre for Travel Medicine, WHO Collaborating Centre for Travellers’ Health, Epidemiology Biostatistics and Prevention Institute, Switzerland.	Should COVID-19 vaccination passports be digital or paper-based? While issues of falsification or counterfeit COVID-19 vaccine certificates surround the paper-based approach, privacy issues also surround the digital approach.	The processes of verifying paper-based and digital COVID-19 vaccine certificates were not analysed. An inclusion of biometric data capturing for issuance of digital COVID-19 vaccination certificate would eliminate physical contact during verification.
2021	Kowalewski et al. [10]	Proof-Of-Vax: Studying User Preferences and Perception of COVID Vaccination Certificates	Ruhr University Bochum, Germany.	A web-based study evaluated user perception of COVID-19 vaccination certificates. Two paper-based variants and three digital app variants were used. Digital and paper-based vaccine certificates were used for documentation purposes only [assumed]. The results showed that paper-based vaccination certificates were generally favoured over digital COVID-19 vaccination certificates.	The assumption that both digital and paper-based vaccination certificates were used for documentation purposes only limited literature discussions.
2021	Mbunge et al. [15]	Emerging technologies and COVID-19 digital vaccination certificates and passports	Department of Information Technology, Faculty of Accounting and Informatics, Durban University of Technology, Durban, South Africa.	Digital COVID-19 vaccination certificates can be used to execute various functions. By applying artificial intelligence, fake COVID-19 vaccination certificates can be easily detected, non-vaccinated areas can be identified and mapped for strategic vaccination planning, and contact tracing can be improved. Furthermore, Internet of Medical Things (IoMT) and blockchain technology can be applied in areas with remote access to COVID-19 vaccination certificates, ensuring data privacy.	To ensure cellular technology does not become a barrier, verification strategies must be designed to require verification to be carried out using the name of the vaccinated person. This will be possible when biometrics of vaccinated people are captured.

**Table 9 vaccines-10-02040-t009:** Addressing issues of privacy concerns for digital COVID-19 vaccine certificates.

Year of Publication	Citation	Title	Study Location	Key Findings	Critique
2021	Hernández-Ramos et al. [7]	Sharing Pandemic Vaccination Certificates through Blockchain: Case Study and Performance Evaluation	European Commission, Joint Research Centre, Ispra, Italy.	Blockchain technology (BC) is a promising approach as a result of its decentralising and transparency features.	The study was silent on how blockchain technologies could be integrated into existing digitalised COVID-19 vaccination certification systems.
2021	Shah et al. [14]	Blockchain for COVID-19: a comprehensive review	Department of Computer Science and Engineering, Institute of Technology, Nirma University, Ahmedabad, Gujarat, India.	BC provides a trustworthy and secured way to share confidential and personal data of patients. It has three key characteristics, namely immutability, transparency and decentralisation.	Public-permissioned blockchain systems redirect data to central servers, making it difficult to apply the decentralised feature that comes with blockchain.
2021	Foy et al. [27]	Blockchain-based governance models for COVID-19 digital health certificates: A legal, technical, ethical and security requirements analysis	Department of Information Technology, Faculty of Accounting and Informatics, Durban University of Technology, Durban, South Africa.	Digital COVID-19 vaccination certificates can be used to execute various functions. By applying artificial intelligence, fake COVID-19 vaccination certificates can be easily detected, non-vaccinated areas can be identified and mapped for strategic vaccination planning, and contact tracing can be improved. Furthermore, Internet of Medical Things (IoMT) and blockchain technology can be applied in areas with remote access to COVID-19 vaccination certificates, ensuring data privacy.	How this limitation can be mitigated to ensure intruders have no access to personal information of vaccinated persons was not highlighted.

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
