# Peer review of "Moving towards Digitising COVID-19 Vaccination Certificate: A Systematic Review of Literature"

_vaccines, 2022, doi:10.3390/vaccines10122040_

Round 1

Reviewer 1 Report

The systematic review of Kissi et al. describes the importance of digital certificates across the countries. The paper is well written, the methods are chosen also well. I have any significant (major) comment, except I see a very low audience for such topic.

I have only a few minor comments:

L45 – Covid-19 please rewrite according to the rest of the document

L130 – Figure 1 has to be reformated, the n number is not visible in several boxes

L160 – trend line in the graph has no significance, I suggest removing it. Also, the legend is not necessary

Table 4: please change the name of the virus according to the nomenclature (SARS-CoV-2 virus, not corona virus)

Author Response

RESPONSE TO REVIEWER 1

Journal:     MDPI - Vaccines

Manuscript #:   vaccines-1787386

Title of Paper: Moving towards digitizing COVID-19 vaccination certificate: A systematic review of literature

Order of Authors: JONATHAN KISSI1*, Emmanuel Kusi Achampong1, Nathan Kumasenu  Mensah1, Caleb Annobil1 and Jessica Naa Lamptey1

Date Sent:      2022.06.27     

Dear Editor

Thank you for your letter and for the reviewers’ comments. We are pleased to resubmit for publication the revised version of our manuscript- Moving towards digitizing COVID-19 vaccination certificate: A systematic review of literature. We appreciate the time, efforts and constructive criticisms by the editors and reviewers. We have addressed all issues indicated, and believe the revised version can meet the journal publication requirements. The revised portions are highlighted in red text in the manuscript. The main corrections in the manuscript and the responses to the reviewers’ comments are as follows:

Comment:  L45 – Covid-19 please rewrite according to the rest of the document

Response: Thank you very much for this comment, we very much appreciate this comment. Authors have addressed this comment and have written Covid-19 as COVID-19 in L46 as suggested.

Comment:  L130 – Figure 1 has to be reformatted, the n number is not visible in several boxes

Response:  Thank you very much for this comment, we very much appreciate this comment. Authors have addressed this comment and reformatted the Figure 1 as suggested.

Comment:  L160 – trend line in the graph has no significance, I suggest removing it. Also, the legend is not necessary

Response:  Thank you very much for this comment, please, authors have corrected the anomaly as advised, the trend line and the legend has been remove from Figure 2

Comment:  Table 4: please change the name of the virus according to the nomenclature (SARS-CoV-2 virus, not corona virus)

Response: Thank you very much for this comment, please, authors have rewritten the nomenclature as advised in Table 4. SARS-CoV-2

Reviewer 2 Report

None, very good, congratulations.

Author Response

RESPONSE TO REVIEWER 2

Journal:     MDPI - Vaccines

Manuscript #:   vaccines-1787386

Title of Paper: Moving towards digitizing COVID-19 vaccination certificate: A systematic review of literature

Order of Authors: JONATHAN KISSI1*, Emmanuel Kusi Achampong1, Nathan Kumasenu  Mensah1, Caleb Annobil1 and Jessica Naa Lamptey1

Date Sent:      2022.06.27     

Dear Reviewer

Thank you for your comments. We are pleased to resubmit for publication the revised version of our manuscript- Moving towards digitizing COVID-19 vaccination certificate: A systematic review of literature. We appreciate the time, efforts and constructive criticisms by the editors and reviewers. We have addressed all issues indicated, and believe the revised version can meet the journal publication requirements.

Reviewer 3 Report

The world-wide spread of the Covid-19 virus and the resulting pandemic have been greatly facilitated by the ease and extent of travel in our society, especially by air.  Once it became apparent in early 2020 that the virus was so easily transmitted and subsequently when the efficacy of rapidly developed vaccines was established, it was obvious that the vaccination status of world travelers was an important consideration for their admittance into a country.  Most countries have adopted a paper-based vaccination card system as verification of one’s vaccination status.  This manuscript summarizes the pros and cons of digitization of the Covid-19 vaccination status of individuals.  Such a digital approach will serve as an easily accessible source that countries all over the world can use to more rapidly identify the vaccination status of travelers with the goal of mitigating the introduction of the virus into their country.  The repeated emergence of new variants of the virus all over the world makes the prevention of their spread into new areas a major goal toward the control of the virus.

This review contributes an interesting, extremely thorough, balanced and informative treatment of the comparison of the use of digital vs. paper-based Cavid-19 vaccine registries.  The authors are to be commended for their fair-handed treatment of the topic, especially in their recognition of the enhanced risk inherent with the digital approach.  Their careful evaluation of the literature and elimination of inappropriate (for several reasons) prospective studies is a strength of the manuscript, as is the potential for abuse of the digital system.  There are only relatively minor criticisms of the manuscript.  

First in lines 38-40 in the introduction, the authors mention that the process of obtaining a vaccination card could help spread the virus, but do not elaborate any further.  By this, I assume they mean that some individuals will fear that they could be tracked by doing so, thus encouraging those individuals no to participate  But, the authors should specify exactly what they mean here.

Second, Table 2 can be eliminated, as the information contained within is very straightforward and is already included in the text.

Third, the term, “block-chain technology” is used throughout the manuscript, but is never defined.  The authors should better explain this term. 

Fourth, the English language usage requires considerable editorial attention.

Author Response

RESPONSE TO REVIEWER 3

Journal:     MDPI - Vaccines

Manuscript #:   vaccines-1787386

Title of Paper: Moving towards digitizing COVID-19 vaccination certificate: A systematic review of literature

Order of Authors: JONATHAN KISSI1*, Emmanuel Kusi Achampong1, Nathan Kumasenu  Mensah1, Caleb Annobil1 and Jessica Naa Lamptey1

Date Sent:      2022.06.27     

Dear Editor

Thank you for your letter and for the reviewers’ comments. We are pleased to resubmit for publication the revised version of our manuscript- Moving towards digitizing COVID-19 vaccination certificate: A systematic review of literature. We appreciate the time, efforts and constructive criticisms by the editors and reviewers. We have addressed all issues indicated, and believe the revised version can meet the journal publication requirements. The revised portions are highlighted in red text in the manuscript. The main corrections in the manuscript and the responses to the reviewers’ comments are as follows:

Comment:  First in lines 38-40 in the introduction, the authors mention that the process of obtaining a vaccination card could help spread the virus, but do not elaborate any further.  By this, I assume they mean that some individuals will fear that they could be tracked by doing so, thus encouraging those individuals no to participate  But, the authors should specify exactly what they mean here.

Response: Thank you for your comment, we are pleased with this comment. Authors have addressed the concerns shown.

Although COVID-19 vaccination certificates can help ease restrictions, the processes involved in obtaining and using it should not be seen as agent of promoting the spread of the virus. Failure to focus on security systems to store patient’s information may raise data privacy concerns that can discourage many from partaking in the vaccination exercise[8] in line 37 to 41.

Comment:  Third, the term, “block-chain technology” is used throughout the manuscript, but is never defined.  The authors should better explain this term. 

Response: Thank you very much for this comment, we very much appreciate this comment. Authors have addressed this comment and have define block-chain technology . Blockchain (BC), which is an immutable, transparent and decentralized technology that provides secured way of sending information may ensure patient’s personal data confidentiality [26] in line 67-69.

Comment:  Fourth, the English language usage requires considerable editorial attention.

Response: Thank you very much for this comment, we very much appreciate this comment. Authors have glanced through the manuscript thoroughly and edited all errors.
